# TWINGEN: protocol for an observational clinical biobank recall and biomarker cohort study to identify Finnish individuals with high risk of Alzheimer's disease

Eero Vuoksimaa [iD],[1] Toni T Saari [iD],[1] Aino Aaltonen,[1] Sari Aaltonen [iD],[1] Sanna-Kaisa Herukka [iD],[2,3] Paula Iso-Markku [iD],[1] Tarja Kokkola [iD],[2] Aija Kyttälä,[4] Sari Kärkkäinen,[2] Hilkka Liedes,[1,5] Miina Ollikainen,[1,6] Teemu Palviainen,[1] Ilona Ruotsalainen,[1,7] Auli Toivola,[4] Mia Urjansson,[1] Tommi Vasankari [iD],[8,9] Henri Vähä-Ypyä,[8] Markus M Forsberg,[10,11] Mikko Hiltunen [iD],[2] Anu Jalanko,[1] Reetta Kälviäinen [iD],[2,3] Teijo Kuopio,[12] Jaakko Lähteenmäki,[7] Pia Nyberg,[13,14] Minna Männikkö [iD],[15] Raisa Serpi,[13] Sanna Siltanen,[16] Aarno Palotie [iD],[1,17,18] Jaakko Kaprio [iD],[1] Heiko Runz [iD],[1,19] Valtteri Julkunen,[2,3] FinnGen

EV and TTS contributed equally.

**Correspondence to**
Dr Toni T Saari;
toni.saari@helsinki.fi

## ABSTRACT

**Introduction** A better understanding of the earliest stages of Alzheimer's disease (AD) could expedite the development or administration of treatments. Large population biobanks hold the promise to identify individuals at an elevated risk of AD and related dementias based on health registry information. Here, we establish the protocol for an observational clinical recall and biomarker study called TWINGEN with the aim to identify individuals at high risk of AD by assessing cognition, health and AD-related biomarkers. Suitable candidates were identified and invited to participate in the new study among THL Biobank donors according to TWINGEN study criteria.

**Methods and analysis** A multi-centre study (n=800) to obtain blood-based biomarkers, telephone-administered and web-based memory and cognitive parameters, questionnaire information on lifestyle, health and psychological factors, and accelerometer data for measures of physical activity, sedentary behaviour and sleep. A subcohort is being asked to participate in an in-person neuropsychological assessment (n=200) and wear an Oura ring (n=50). All participants in the TWINGEN study have genome-wide genotyping data and up to 48 years of follow-up data from the population-based older Finnish Twin Cohort (FTC) study of the University of Helsinki. The data collected in TWINGEN will be returned to THL Biobank from where it can later be requested for other biobank studies such as FinnGen that supported TWINGEN.

**Ethics and dissemination** This recall study consists of FTC/THL Biobank/FinnGen participants whose data were acquired in accordance with the Finnish Biobank Act. The recruitment protocols followed the biobank protocols approved by Finnish Medicines Agency. The TWINGEN study plan was approved by the Ethics Committee

## STRENGTHS AND LIMITATIONS OF THIS STUDY

⇒ A large sample of individuals is recruited from a representative biobank database.

⇒ Using health registry information, we exclude those with documented Alzheimer's disease (AD) or other neurological or psychiatric diseases that can affect cognition. Prescreening limits the sending of unnecessary invitations and saves costs.

⇒ Participants have up to 48 years of follow-up questionnaire and clinical data from the Finnish Twin Cohort study and these data can be combined with multifaceted Finnish health registry information. Previous genotype data is available in the biobank from all TWINGEN study participants.

⇒ We assess the feasibility of remote cognitive testing and blood samples in large-scale screening of AD risk, translating to the requirements of intervention trials and clinical practice.

⇒ Limitations of the study are a lack of gold standard biomarkers (cerebrospinal fluid, positron emission tomography imaging) and neurological examinations.

of Hospital District of Helsinki and Uusimaa (number 16831/2022). THL Biobank approved the research plan with the permission no: THLBB2022_83.

## INTRODUCTION

Alzheimer's disease (AD)—the most common cause of dementia—is characterised by pathological accumulation of beta-amyloid (Aβ) and tau in the brain.[1] As populations age,

the prevalence of dementia is projected to nearly double every two decades[1 2] and AD and related dementias are becoming one of the most common causes of death in many countries (20% of deaths in Finland ranking it as the third most common cause of death).[3] The AD process starts up to 20–30 years before the diagnosis, so intervention trials targeted at preclinical or prodromal stages of AD are of high priority, but time-consuming and costly with screen-failure rates of 78%–88%.[4]

In clinical practice, AD is diagnosed mainly based on the clinical phenotype, episodic memory impairment being the cognitive hallmark, while in research there has been a shift from clinical diagnosis to biological classification independent of the cognitive status.[5] Evidence of diagnostic properties of different blood-based biomarkers is rapidly accumulating,[6] but population-based studies are still scarce. In one population-based study, 11% of older adults (median age 74 years) without dementia were found to have Aβ pathology using blood-based biomarkers.[7]

Our earlier pilot study explored the feasibility of FinnGen, a nationwide Finnish biobank study[8] in recruiting individuals with AD to an observational study via the biobank.[9] Our protocol included blood-based biomarkers and remote cognitive assessment, approaches that are suggested to improve the recruitment of participants for AD trials.[10–12] Another FinnGen study showed that biobank participants could be recontacted for additional data collection on a larger scale as well.[13] The full potential of population biobank datasets lies in the large cohorts of undiagnosed individuals who are at the risk of developing AD in near future and might be suitable for targeted screening for early diagnostics and interventional trials encompassing both pharmacological treatments and lifestyle interventions.

After showing in our pilot study that recall of biobank participants with AD to a clinical study including multimodal (remote and in-person) cognitive assessment and blood-based biomarker analyses is feasible,[9] we modified the protocol of the pilot study to target cognitively unimpaired older adults (based on health registry data) in the current study. Additionally, we augmented the assessment battery with passive technology for measuring physical activity, sedentary behaviour and sleep.

This protocol paper describes TWINGEN, a population-based follow-up study investigating the utility of easily implementable methods for assessing the risk of AD. We aim to conduct a proof-of-principle study for using biobank registries as a platform for recruiting participants suitable for clinical trials, particularly in diseases where recruitment and screening have generally been challenging. We also focus on the remote cognitive assessment methods and blood-based biomarkers of AD. The study also aims to enrich existing biobank data derived from a long-standing prospective twin study with cognitive and lifestyle measures. The research setting is unique as it uses the prescreening and recall option based on the data of the biobank in combination with long preceding population-based follow-up data from the twin study.

## METHODS AND ANALYSIS
### Study participant selection
The target group of the TWINGEN study are individuals who have participated in the older Finnish Twin Cohort (FTC) study of the University of Helsinki (UH), and whose samples and data have been transferred to THL Biobank in 2018. The FTC was chosen as the primary target of this study because it is a population-based follow-up study with up to 48 years of previous comprehensive health data available. Combining the historical data with newly collected samples would allow building of longitudinal trajectories of various lifestyle and health factors to late-life cognitive decline. The main selection criteria in the biobank were previous participation in FTC, age (65–85), place of current residence in Finland, Finnish as the first language and no known diagnosis affecting cognition in biobank records. The selection of eligible FTC study participants was done through THL Biobank and data collection was carried out by the UH for those living in the greater Helsinki area or surrounding regions and by regional biobanks and Turku University of Applied Science based on the residency of the participant. Selection also included participation in the FinnGen nationwide biobank research study for two reasons.[8] First, FinnGen supported the collection of the TWINGEN cohort with the aim to enrich phenotype information. Second, the collected TWINGEN data are to be returned to THL Biobank, from where they can later be requested for the FinnGen study and combined with its extensive gene and health register data. Thus far, FinnGen has produced genotype data from ca. 500 000 biobank donors of all Finnish biobanks to perform large-scale genome and health research. In the following sections, we describe each of the data sources and the study protocol of TWINGEN.

### The older FTC study
The older FTC study from the UH is a population-based study that includes all Finnish same-sex twins born before 1958 and living in Finland at the start of the study in 1974 (figure 1).[14] The baseline survey was conducted in 1975 via postal questionnaire and 27 750 individuals participated with an 89% participation rate. Follow-up questionnaires were sent in 1981 (n=24 684 with an 84% response rate), 1990 for those born in 1930 or later (n=12 502; 77%) and in 2011–2012 for those born in 1945–1957 (n=8410; 72%).[15] Those born in 1938–1944 have also participated in MEMTWIN II study (n=1772) that used telephone interview to assess cognition.[16] Some of the twins born in 1945–1957 have participated in Essential Hypertension EPIgenetics study (EH-EPI, n=445).[17] The MEMTWIN II and EH-EPI study participants are the primary groups of interest in TWINGEN because they have either earlier

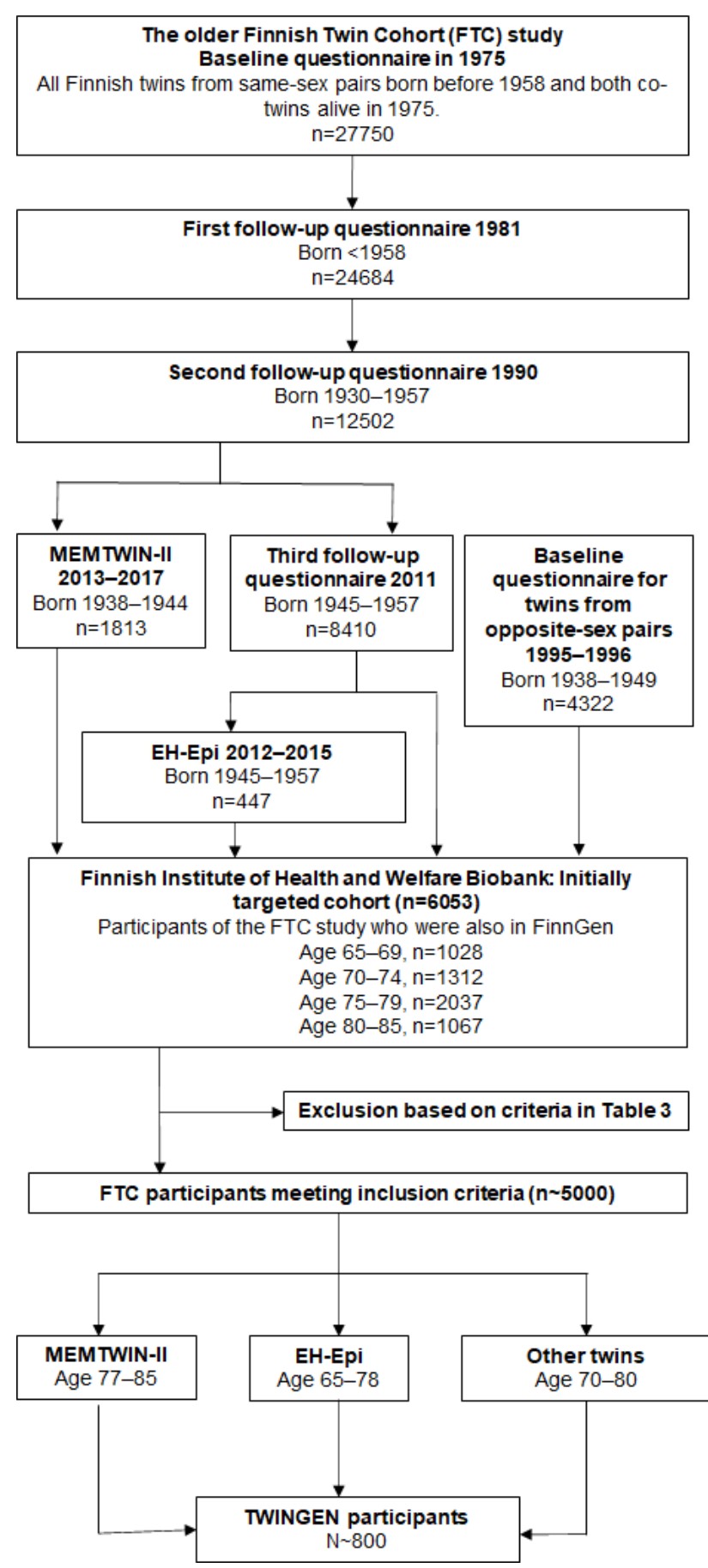

**Figure 1** Flowchart of study participant selection. EH-Epi, Essential Hypertension EPIgenetics.

**Table 1** Cognitive measures used in the TWINGEN study

| In-person neuropsychological assessment | Telephone (TELE/TICS-m3) | Web-based computerised assessment (cCOG) |
|---|---|---|
| **Memory and learning** | | |
| CERAD Word List Learning | Word List Learning | Episodic Memory Learning |
| CERAD Word List Recall | Word List Recall | Episodic Memory Recall |
| CERAD Word List Recognition | | Episodic Memory Recognition |
| CERAD Constructional Praxis Recall | | |
| WMS-III Logical Memory Story A | | |
| WMS-III Logical Memory Story A Recall | | |
| **Executive function** | | |
| Trail Making Test-B | | Modified Trail Making Test-B |
| Stroop Interference | | |
| Stroop Set-shifting | | |
| **Visuospatial skills and visuoconstruction** | | |
| CERAD Constructional Praxis | | Fragmented Letters |
| CERAD Clock Drawing Test | | |
| **Language skills and fluency** | | |
| CERAD Naming Test | Similarities | |
| CERAD Semantic Fluency | Semantic Fluency | |
| **Processing speed** | | |
| Trail Making Test-A | | Modified Trail Making Test-A |
| Stroop Word Reading | | |
| Stroop Colour Naming | | |
| **Global cognition** | | |
| CERAD MMSE | TELE Global Score | cCOG Global Score |
| | TICS Global Score | |
| | TICS-m3 Global Score | |

MEMTWIN-II participants have prior TELE/TICS-m data.
CERAD-nb, Consortium to Establish a Registry for Alzheimer's Disease-neuropsychological battery; MMSE, Mini-Mental State Examination; TELE, telephone assessment for dementia; TICS, Telephone Interview for Cognitive Status; TICS-m, modified Telephone Interview for Cognitive Status; TICS-m3, TICS m including three learning trials in the Word List Learning; WMS-III, Wechsler Memory Scale 3rd edition.

cognitive (MEMTWIN II; table 1) or multi-omics data (EH-EPI; online supplemental table 1) available.

To achieve the target number of 800 participants, the biobank selection was expanded to the twins born in 1945–1952 who had not participated in MEMTWIN II

or EH-EPI. Most of the invited twins were from same-sex pairs, but we also invited twins from opposite-sex pairs included in the older FTC study in year 1995–1996, when they replied to a brief health questionnaire (figure 1). We prioritised invitations to participants who lived closest to one of the six study sites. All TWINGEN participants from same-sex twin pairs have longitudinal questionnaire data on health and health-related behaviours from years 1975, 1981 and 1990, and those born in 1945–1957 have data also from year 2011 to 2012. An overview of longitudinal data available for the MEMTWIN II and EH-EPI and other TWINGEN participants are presented in table 2. Necessary inclusion criteria for TWINGEN were available DNA sample in THL Biobank and not having any of the exclusion criteria (table 3). Additional references for studies using the older FTC data are found in online supplemental table 2.

## Recall procedure

The results of a THL Biobank feasibility assessment indicated that there were 6053 individuals aged 65–85 who have participated in the FTC of UH and have data in THL Biobank. Approximately 1000 individuals were excluded due to our exclusion criteria of AD and other neurological or psychiatric diseases that can affect cognition (figure 1; table 3). Furthermore, the genotype data of approximately 100 individuals have been verified by FinnGen, but these data have not been returned to their respective biobanks at the time of the feasibility assessment; thus, these individuals are excluded from the pool of potentially eligible participants. The target sample size of 800 participants was chosen as it fits the timeframe and resources of the study and is sufficiently large for stratifying participants at varying risks for AD.[7]

As a biobank recall study, TWINGEN participants are contacted by THL Biobank by an invitation letter. The invitation letter includes information about the individuals' prior participation to the older FTC study and the transfer to and storage of samples and data at THL Biobank, information about the participation in the FinnGen study via the biobank, and information about the new TWINGEN study. The invitation letter also contains two separate consent forms: one for the participation in the TWINGEN study and the biobank consent for THL Biobank. The consent for THL Biobank is needed to store the new samples and data obtained in TWINGEN to THL Biobank and to confirm the biobank participation with a written biobank consent, which is the primary basis for storing samples and data into a biobank. After receiving consent forms, UH research staff contacts the potential participants to verify that the individuals have understood the purpose and procedures of the study. Additionally, as the health registry data available in the biobank is not up to date but reflects status at the end of year 2022, research staff verifies (via telephone) that the exclusion criteria are not met. Eligibility of each individual is assessed independently of their co-twin's vital status or eligibility.

**Table 2** Previous data from the participants recruited in TWINGEN

| | Baseline assessment (1975) | First follow-up (1981) | Second follow-up (1990) | Third follow-up (2011–2012) | MEMTWIN-II (2013–2017)* | EH-Epi (2012–2015) | Primary study reference |
|---|---|---|---|---|---|---|---|
| Education | × | × | | | | | Vuoksimaa et al, 2016b |
| Chronic or serious illness | × | × | × | × | × | × | Kaprio et al, 2019[14] |
| Physical activity | × | × | × | × | | × | Piirtola et al, 2017 |
| Smoking | × | × | × | × | | × | Kaprio and Koskenvuo 1988 |
| Alcohol use | × | × | × | × | | × | Sipilä et al, 2016 |
| Sleep | × | × | × | × | | | Kaprio et al, 2019[14] |
| Medications | × | × | × | × | × | × | Huang et al, 2018 |
| Anthropometrics (weight, height) | × | × | × | × | | × | Piirtola et al, 2017 |
| Dietary habits | | × | | | | × | Kaprio et al, 2019[14] |
| Blood pressure | × | × | × | × | | × | Iso-Markku et al, 2021 |
| Cholesterol | | × | × | × | | | Iso-Markku et al, 2021 |
| Diabetes | × | × | × | × | | × | Iso-Markku et al, 2021 |
| Subjective memory complaints | | | | | × | | |
| Life satisfaction | × | × | × | × | | | Koivumaa-Honkanen et al, 2000 |
| Loneliness | × | × | × | × | | | Koivumaa-Honkanen et al, 2000 |
| Social support | | | × | × | | | Romanov et al, 2003 |
| Cognition (TELE/TICS-m) | | | | | × | | Lindgren et al, 2019[16] |
| Depressive symptoms | | | BDI | CES-D | CES-D | | Saari et al, 2023[46] |
| Personality (EPI) | × | × | | × | | | Rose et al, 1988[50] |
| Accelerometer (physical activity) | | | | | × | | Waller et al, 2019 |
| Multi-omics data | | | | | | × | Drouard et al, 2022 |
| DNA | × | × | × | × | × | × | Kaprio et al, 2019[14] |

See online supplemental file 3 for full reference information and additional references on previous studies using older Finnish Twin Cohort study data. *Third follow-up assessments for those born in 1938–1944.
BDI, Beck Depression Inventory; CES-D, Center for Epidemiological Studies-Depression scale; EPI, Eysenck Personality Inventory; LS, life satisfaction; TELE, telephone assessment for dementia; TICS-m, modified Telephone Interview for Cognitive Status.

**Table 3** Exclusion criteria for the TWINGEN study

| ICD-10 code | Explanation |
|---|---|
| G30 and F00 | Any variant of Alzheimer's disease or any dementia relating to Alzheimer's disease, dementia with Lewy bodies, frontotemporal dementia and mixed dementia |
| F01-F03 | Dementia, any aetiology |
| G20 | Parkinson's disease |
| G35 | Multiple sclerosis |
| I60, I61 and I63 | Intracerebral haemorrhage, subarachnoid haemorrhage or ischaemic stroke and their subcategories |
| S06.1–S06.7 | Traumatic brain injuries other than concussion (S06.0) |
| F20 | Schizophrenia |
| F31 | Bipolar disorder |
| F33, F32.1–F32.3, F34 | Recurrent depression, moderate or severe depression and long-lasting mood disorders |
| F60 | Personality disorders |
| F10 | All diagnoses relating to excessive usage of alcohol |
| F11–F19 | Intoxication because of opioids, cannabinoids, sedative medication, cocaine or hallucinogens |
| F70–F73 | Different stages of intellectual disabilities |

## Data collection

Data collection started on 29 March 2023 and is planned to be completed by the end of 2023. Data collection is conducted in six locations across Finland: at the Institute for Molecular Medicine Finland (FIMM), UH in Helsinki, biobanks in four locations across Finland: in Jyväskylä (Central Finland biobank), Kuopio (Biobank of Eastern Finland), Oulu (Arctic Biobank and Biobank Borealis of Northern Finland) and Tampere (Finnish Clinical Biobank Tampere) and at the clinical laboratory of Turku University of Applied Sciences. Protocol for all participants includes telephone-administered and computer-administered cognitive testing, blood-draw and self-report questionnaire. Additionally, a waist-worn accelerometer will be given to participants who are willing to complete 1 week at-home measurement. Furthermore, individuals living in the greater Helsinki area are invited to participate in in-person neuropsychological testing (target sample size n=200) and measurement of weight, height, waist circumference and blood pressure. In Helsinki, we also provide Oura rings for 50 participants who are willing and able to wear the ring and use the associated mobile app. After participating in the study, participants receive a report based on their performance in telephone interview-based and computerised tests of cognition and about their physical activity. Data cleaning, feedback to participants and transfer of data to THL biobank are estimated to be completed by the end of June 2024. A detailed description of the data collections is provided later.

### Telephone interview for assessing cognitive status and function

Two validated telephone-administered cognitive screening instruments are used: a telephone assessment for dementia (TELE)[18] and the modified Telephone Interview for Cognitive Status (TICS-m).[19] TELE and TICS have been translated to Finnish and adapted to Finnish culture[20] and both have been used in the older FTC study. TICS-m, a modified version of TICS includes an additional delayed free recall of the 10-word list and has been used in the MEMTWIN II substudy.[21] We further modified the TICS-m by including three learning trials of the 10-word list and this instrument is later referred to as TICS-m3.[9] We also included semantic fluency. Three trial word list task yields immediate and delayed free recall measures of episodic memory. Semantic fluency score is the number of animals named in 1 min (table 1). In the telephone interview, the participants are also asked about their functional abilities regarding household maintenance, ambulation, shopping, dressing and undressing, use of mobility aids, memory problems and possible visits to the doctor regarding memory problems. Telephone interviews were conducted by trained study nurses or psychologists.

### Computerised web-based cognitive testing

Web-based cCOG tool created by Combinostics (Tampere, Finland) is used for computerised cognitive testing.[22] It includes six subtests: Episodic Memory (learning and recall), Reaction Time, Modified Trail Making A and B and Distorted Letters. The tasks measure visual processing, memory, processing speed, attention and executive function (table 1). In addition to cognitive tests, cCOG includes background questions (education) and a 7-item questionnaire designed to assess probable dementia with Lewy bodies.[23] The test battery takes about 25 min to complete and is performed via a keyboard and a mouse or a touchscreen device.

### In-person neuropsychological tests

The participants in Helsinki study site undergo an in-person neuropsychological assessment, with a target sample size of 200 individuals. The minimum sample size to detect medium correlations (r=0.3) between cognitive measures with a significance level=0.05 and power=0.8 is n=85, thus, the target sample size of 200 individuals with in-person cognitive assessment is adequate for examining correlations between in-person and remote cognitive measures. The larger sample size is expected to be adequate for factor analyses of the neuropsychological battery.

The in-person neuropsychological assessment consists of the Consortium to Establish a Registry for Alzheimer's Disease-neuropsychological battery (CERAD-nb)[24] and tests measuring executive functions, processing speed and episodic memory. The CERAD-nb includes

Mini-Mental State Examination and Semantic Fluency, abbreviated Boston Naming Test, Word List Learning, Recall and Recognition, and Constructional Praxis (Copy and Recall). The Finnish version of the CERAD-nb also includes Clock Drawing test.[25] Finnish education adjusted cut-offs are available for total score and for each subtest.[26] In addition to the CERAD-nb, we include the following tests: Logical Memory Story A from the Wechsler Memory Scale 3rd edition,[27] Trail Making Test A and B[28] and Stroop test.[29] The version of Stroop used in this study is the 40-item version used in the FINGER study[30] with an additional fourth 'set-shifting' condition whereby the task is to name the colour of ink (as in the classical Stroop condition) or to read out the colour-word when the word is inside a rectangle. Neuropsychological tests are administered by trained psychologists. A summary of key cognitive measures of all three modalities are presented in table 1.

### Blood sample

A venous blood sample is drawn from the inside of the elbow or alternatively from the back of the hand. A total of six tubes are collected: three BD Vacutainer K2EDTA (10/10 mL) tubes, two BD Vacutainer SSTII Advance-serum gel tubes (10/8.5 mL) and one BD PAXgene Blood RNA (7/2.5 mL) tube (online supplemental figure 1). Processing will be done immediately after the samples have been taken. Serum tubes are allowed to clot at least 30 min (max. 60 min) before separating. EDTA-plasma tubes do not clot but two EDTA tubes are also let sit for the same time as serum because of the easier workflow. Serum and two EDTA-plasma tubes are centrifuged (1500g) for 10 min and the supernatants are pooled within serum and plasma. Samples are apportioned into 0.5 mL aliquots and stored at −20°C. One EDTA tube and PAXgene RNA-tube will be stored as a whole blood at −20°C. Serum, plasma and RNA samples final storage temperature is at −80°C. One of the 0.5 mL EDTA-plasma aliquot will be sent to University of Eastern Finland for biomarker analysis. RNA, EDTA whole blood samples, and half of the serum and plasma aliquots are dedicated to the twin study, while the other half will be available for research via THL Biobank.

### Blood-based AD biomarkers

The primary blood-based biomarkers include phosphorylated-tau181 (p-tau181), phosphorylated-tau217 (p-tau217), Aβ1-42/40, glial fibrillary acidic protein (GFAP) and neurofilament light chain (NfL); all measured using Simoa HD-X Analyzer (Quanterix, Billerica, Massachusetts, USA). Plasma p-tau181 levels are quantified using Simoa p-tau181 Advantage V2.1 Kit (Ref# 104111, Quanterix),[31] Aβ1–40, Aβ1–42, GFAP and NfL levels using Simoa Neurology 4-Plex E Advantage Kit (Ref# 103670, Quanterix),[32] and p-tau217 levels using ALZpath Simoa pTau-217 v2 Assay Kit (Ref# 104371, Quanterix).[33] Prior to analyses, EDTA plasma samples are thawed, mixed and centrifuged (10 000×g, 5 min, +20°C).

**Table 4** Core UKK RM42 accelerometer and Oura ring parameters

| UKK RM42 accelerometer | Oura ring |
|---|---|
| Target n=800 | Target n=50 |
| Participants from all sites | Participants from Helsinki site |
| Wearing time=1 week | Wearing time=2 weeks |
| Physical activity parameters | |
| Light physical activity | Low intensity activity |
| Moderate physical activity | Medium intensity activity |
| Vigorous physical activity | High intensity activity |
| Total physical activity | Total physical activity |
| Number of steps | Number of steps |
| Standing time | |
| Sedentary behaviour parameters | |
| Lying time | Inactive time |
| Reclining time | Resting time |
| Sitting time | |
| Number of breaks during sedentary time | |
| Sleep parameters | |
| Total sleep time | Total sleep time |
| Restless sleep time | Restless sleep percentage |
| Restful sleep time | Total amount of deep sleep |
| | Total amount of rapid eye movement sleep |
| | Total amount of light sleep |

These biomarkers are determined at the Biomarker Laboratory of University of Eastern Finland.

### Apolipoprotein E genotype and polygenic risk scores

Apolipoprotein E (APOE) status (ε4-carrier vs non-carrier and number of ε4-alleles) is defined by two single-nucleotide polymorphisms, rs429358 and rs7412, in chromosome 19[34] and polygenic risk score (PRS; with and without APOE) of AD is based on the Bellenguez *et al*[35] or newer meta-analysis if available. Genetic data will be used to calculate also PRS's for diseases and traits (such as cardiovascular disease and educational attainment) that are related to risk and protective factors of dementia.[36]

### Accelerometer-measured physical activity, sedentary behaviour and sleep

A tri-axial accelerometer (UKK RM42, UKK Terveyspalvelut Oy, Tampere, Finland) is used to monitor participants' daily physical activity, sedentary behaviour and sleep for 7 consecutive days (table 4).[37] In addition to total time spent in physical activity of different intensities and sedentary behaviour, we will also measure number of bouts and length of the bouts of physical activity and sedentary behaviour. The participants receive the devices during their in-person visit at FIMM or by mail if participating in other location. Participants are asked to wear the accelerometer on the hip during waking hours and on the

wrist during sleep. At least a 4-day monitoring period with a minimum of 10 hours wear-time a day will be required for the adequate accelerometer data collection.[38]

The UKK RM42 device and its closely related counterpart Hookie AM 20 accelerometer have been used in samples with over 18 000 Finnish 18–85 years old adults.[37 39] Thus, the UKK RM42 accelerometer is usable in the TWINGEN sample of 65–85 years old, and we will get an opportunity to compare the measurements against normative data of Finnish adults. The analyses of raw acceleration data of the UKK RM42 are based on validated algorithms; the technical details related to the recording and analysing of raw acceleration data are given elsewhere.[40–42]

### Oura-measured physical activity, sedentary behaviour and sleep

Oura ring (Gen3 Heritage, Ōura Health Ltd., Oulu, Finland) will be given to 50 participants in Helsinki study collection site. The number of participants asked to wear an Oura ring was an experimental pilot within our larger study. The target sample of 50 was purposed to evaluate the feasibility of a measurement requiring a smart phone app in older adults. First, participants use a ring-size kit to determine optimal ring size and then they receive the ring either at their in-person visit or by mail. Participants are asked to wear the ring (width: 7.9 mm, thickness: 2.55 mm, weight: 4–6 g) in any finger for 2 weeks during day and night, except when charging the ring every 4–6 days (20–80 min to fully charge). To monitor participant's sleep, sedentary behaviour and physical activity (table 4), the Oura ring uses infrared photoplethysmography sensors, negative temperature coefficient sensor and 3D accelerometer. Participants receive written instructions on using the Oura ring and downloading the Oura mobile application, through which they can access their own data. If necessary, the research staff provides phone guidance for both using the ring and installing the application. The data from Oura ring is transferred to participant's Oura application when opening the application and to a cloud server. Data collection will be monitored from Oura cloud server and participants' will be sent a reminder if there are no data from previous 2 days.

The Oura ring's sleep stage detection algorithm (wake, light non-rapid eye movement (NREM) sleep, deep NREM sleep, rapid eye movement sleep) has been validated against polysomnography and it showed 80%–96% accuracy, 74%–82% sensitivity and 79%–98% specificity.[43] Furthermore, moderate-to-vigorous intensity physical activity and step count of Oura ring has shown strong correlations with accelerometer-measured corresponding values.[44]

### Questionnaire

Participants are given a 16-page self-report questionnaire that includes many of the same measures as in previous questionnaires in years 1975, 1981, 1990 and 2011 (table 2). Questions cover anthropometrics, demographics, social relationships, chronotype, health (general, cardiovascular, dementia, memory, medications,

vision, hearing, balance and mobility) and health-related behaviour including sleep, physical activity, smoking and alcohol use. Psychological well-being scales included in the questionnaire are: 8-item Center for Epidemiologic Studies Depression scale[45 46]; 7-item Purpose in Life subscale from Ryff's Scales of Psychological Well-Being[47 48]; Extraversion (9 items) and Neuroticism (10 items) from the short version of Eysenck Personality Inventory[49 50]; and four-item life satisfaction scale derived from questionnaires by Allardt.[51 52]

### Patient and public involvement statement

None.

### Aims, data analysis and future directions

In addition to the overarching aim of assessing the feasibility of biobank recall in the context of preclinical AD, we also have more focused research questions (see online supplemental figure 2 for an overview of aims and associated data). In the initial stage of TWINGEN, we cannot make clinical or research diagnoses of AD, however, we will be able to follow these individuals through national health registry information that are compiled in FinnGen. Registry-based data will allow to predict progression to AD. Possible follow-up visits can also include gold standard measures for diagnosing AD, such as cerebrospinal fluid or positron emission tomography imaging.

By combining all data, we aim to stratify our participants in subgroups of low, intermediate and high AD risk based on genetic, biomarker, cognitive, lifestyle and symptom data. The stratification will be based on a combination of percentiles or cut-offs (eg, Ashton *et al*[33]) for blood-based biomarkers, cut-offs for cognitive impairment in CERAD,[26] cCOG[22] and TELE/TICS,[16] lifestyle risk scores and subjective memory complaints. We also aim to use neuropsychological criteria for mild cognitive impairment classification where −1 SD performance in at least two tests are required independent of subjective memory complaints.[53 54] Biomarkers and cognitive data allow to derive subgroups based on biomarker profile and cognitive stating included in the AT(N) framework.[5] Additionally, we will use APOE status and PRS for genetic risk profiling although genetics are not included in the AT(N) framework.

The stratification approaches are potentially useful for improving participant selection for AD drug and intervention trials. These methods would also be valuable in clinical settings where non-invasive and widely available tools for evaluating the presence of AD pathology underlying cognitive symptoms is important, especially once disease-modifying treatments become available. TWINGEN will also establish a baseline cohort that can be used in follow-up studies with neuroimaging and cerebrospinal fluid biomarkers.

We aim to assess the comparability of in-person, telephone-based and computerised cognitive assessments using correlation analysis of total scores and tests of different modalities assessing the same cognitive

domains (eg, memory). We will also explore the distributions and correlations among blood-based biomarkers of AD-related pathologies and investigate the associations between cognition and biomarkers. Cognition will be treated both categorically (cognitive status) and as continuous outcomes for domain-specific measures (eg, episodic memory, executive function). For the domain-specific cognitive composites, factor scores will be calculated similar to previous studies on preclinical AD and mild cognitive impairment.[55 56] The factor scores will be based on exploratory factor analyses of the in-person neuropsychological battery and used instead of individual tests when the interest is on a cognitive domain, not a single test.

Additionally, the effect of genetic and lifestyle (with up to 48 years of follow-up) factors can be used in predicting cognitive and biomarker status. The measures of physical activity, sedentary behaviour and sleep (table 4) allow us to explore the relationships between physical activity and sleep with cognition and biomarkers. The list of physical activity and sleep parameters in table 4 is not exhaustive and more parameters are available for detailed analyses.

Although inclusion in the study was not dependent on the co-twin's participation, it is expected that full twin pairs will also participate. This will allow studying if the between-family associations are also evident in within-family comparisons and we can identify twin pairs who are discordant for cognition or biomarkers.[57]

## ETHICS AND DISSEMINATION

According to the Finnish Biobank Act, research collections, which have been collected, or whose collection started before the Biobank Act became into force on 1 September 2013, can be transferred to a biobank by a specific procedure, which includes an ethical evaluation and informing the sample donors either personally or by public announcement. Accordingly, biological samples of the FTC, including DNA and associated data were transferred to THL biobank in December 2018 to facilitate biobank research. The action of transferring FTC data to THL Biobank was publicly announced in major newspapers.

Recruitment protocols followed the biobank protocols approved by Fimea. The FinnGen study was approved by the Coordinating Ethics Committee of Hospital District of Helsinki and Uusimaa (HUS; statement number HUS/990/2017). The permit numbers of the decisions made by Finnish Institute for Health and Welfare (THL) and the Biobank Access Decisions for FinnGen samples and data used in FinnGen Data Freeze 9 are presented in the Acknowledgements section.

Before entering the TWINGEN study all potential study participants received detailed information regarding this study by a formal information letter. All study participants were asked for and provided written informed consent. The recall study was reviewed and approved by the ethics committee of HUS (approval number 16831/2022). THL Biobank approved the research plan with the permission no: THLBB2022_83.

The data acquired in this study are managed and initially stored by UH. Keeping with the Biobank Act and the consent given by the participants, the data are also transferred to THL Biobank for later research use. FinnGen request to use the TWINGEN data from THL Biobank, after which the data acquired in this study are linked with existing genetic and register data in a controlled FinnGen sandbox environment.

The data acquired in this study is subject to conditions of the IRB protocols and the policies of the Finnish biobank legislation and therefore unavailable for unsupervised usage. The data is stored in FIMM and THL Biobank, where approved researchers can access the data. Eventually the data will also be transferred to the secure FinnGen sandbox environment and linked to the registers available in FinnGen. The results of the study will be published in peer-reviewed journals and presented at scientific conferences.

## DISCUSSION

The TWINGEN study uses biobank registries for a recall study with the aim of identifying individuals at risk of AD. The collected data comprise many known risk factors[36] and scalable screening methods of AD. While these data are expected to yield novel insights even in isolation, the unique potential for discovery comes from combining the existing follow-up data with the newly collected data of TWINGEN participants. Although this study has many strengths, a limitation is the lack of gold standard biomarkers (cerebrospinal fluid, positron emission tomography imaging) and neurological examinations in our baseline assessment.

Since its inception, the FinnGen project has created new expansion areas with the aim of enriching phenotype information. The TWINGEN study addresses this goal by collecting cognitive, physical activity, lifestyle and biological data that are returned to FinnGen via THL biobank, whereas FinnGen would allow for a registry-based follow-up of the TWINGEN study participants.

### Author affiliations
[1]Institute for Molecular Medicine Finland, University of Helsinki, Helsinki, Finland
[2]Department of Neurology, Institute of Clinical Medicine, University of Eastern Finland, Kuopio, Finland
[3]Department of Neurology, NeuroCenter, Kuopio University Hospital, Kuopio, Finland
[4]THL Biobank, Finnish Institute for Health and Welfare, Helsinki, Finland
[5]VTT Technical Research Centre of Finland Ltd, Oulu, Finland
[6]Minerva Foundation Institute for Medical Research, Helsinki, Finland
[7]VTT Technical Research Centre of Finland Ltd, Espoo, Finland
[8]UKK Institute for Health Promotion Research, Tampere, Pirkanmaa, Finland
[9]Faculty of Medicine and Health Technology, Tampere University, Tampere, Finland
[10]School of Pharmacy, Faculty of Health Sciences, University of Eastern Finland, Kuopio, Finland
[11]VTT Technical Research Centre of Finland Ltd, Kuopio, Finland
[12]Central Finland Biobank, Wellbeing Services County of Central Finland and University of Jyväskylä, Jyväskylä, Finland
[13]Biobank Borealis of Northern Finland, Oulu University Hospital, Wellbeing Services County of North Ostrobothnia, Oulu, Finland

[14]Translational Medicine Research Unit, University of Oulu, Oulu, Finland

[15]Arctic Biobank, Infrastructure for Population Studies, Faculty of Medicine, University of Oulu, Oulu, Finland

[16]Finnish Clinical Biobank Tampere, Tampere University Hospital, Wellbeing Services County of Pirkanmaa, Tampere, Finland

[17]Analytic and Translational Genetics Unit, Department of Medicine, Department of Neurology and Department of Psychiatry, Massachusetts General Hospital, Boston, Massachusetts, USA

[18]The Stanley Center for Psychiatric Research and Program in Medical and Population Genetics, The Broad Institute of MIT and Harvard, Cambridge, Massachusetts, USA

[19]Translational Sciences, Biogen Inc, Cambridge, Massachusetts, USA

**Acknowledgements** We thank THL Biobank and biobank directors Eero Punkka (Helsinki Biobank) and Veli-Matti Kosma (Biobank of Eastern Finland) for providing resources to carry out this study. Biobank personnel who have collected the data and processed the samples: Sabrina Belgasem (Helsinki Biobank), Henna Palin, Minttu Virolainen and Anna-Kaisa Pohjonen (Finnish Clinical Biobank Tampere), Anu Outinen-Tuuponen, Marja-Leena Kytökangas, Riikka-Mari Siiro-Virtanen (Arctic Biobank and Biobank Borealis of Northern Finland), Senni Lipponen (Central Finland Biobank), Nina Hurula (Biobank of Eastern Finland) and Heidi Kalve, Anniina Friman and biomedical laboratory scientist students from the Turku University of Applied Sciences' clinical laboratory. The participants of the TWINGEN study were recruited through THL Biobank (study number THLBB2022_83) and we thank all study participants for their generous participation in biobank research. We thank Jyrki Tammerluoto and Steffi Besselink for legal services and Huei-Yi Shen for administrative work. We thank the participants of the Finnish Twin Cohort study for their participation to TWINGEN and all previous data collection waves.

**Collaborators** FinnGen: Adam Platt (Astra Zeneca, Cambridge, United Kingdom), Adam Ziemann (Abbvie, Chicago, IL, United States), Adriana Huertas-Vazquez (Merck, Kenilworth, NJ, United States), Aino Salminen (Hospital District of Helsinki and Uusimaa, Helsinki, Finland), Airi Jussila (Pirkanmaa Hospital District, Tampere, Finland), Aki Havulinna (Institute for Molecular Medicine Finland (FIMM), HiLIFE, University of Helsinki, Helsinki, Finland; Finnish Institute for Health and Welfare (THL), Helsinki, Finland), Alessandro Porello (Janssen Research & Development, LLC, Spring House, PA, United States), Ali Abbasi (Abbvie, Chicago, IL, United States), Amanda Elliott (Institute for Molecular Medicine Finland (FIMM), HiLIFE, University of Helsinki, Helsinki, Finland; Broad Institute, Cambridge, MA, USA and Massachusetts General Hospital, Boston, MA, USA), Amy Hart (Janssen Research & Development, LLC, Spring House, PA, United States), Anastasia Kytölä (Institute for Molecular Medicine Finland (FIMM), HiLIFE, University of Helsinki, Helsinki, Finland), Anders Mälarstig (Pfizer, New York, NY, United States), Andrea Ganna (Institute for Molecular Medicine Finland (FIMM), HiLIFE, University of Helsinki, Helsinki, Finland), Andrey Loboda (Merck, Kenilworth, NJ, United States), Anne Lehtonen (Abbvie, Chicago, IL, United States), Anne Pitkäranta (Helsinki Biobank / Helsinki University and Hospital District of Helsinki and Uusimaa, Helsinki), Anne Remes (Clinical Neurosciences, University of Helsinki, Helsinki, Finland), Annika Auranen (Pirkanmaa Hospital District, Tampere, Finland), Antti Aarnisalo (Hospital District of Helsinki and Uusimaa, Helsinki, Finland), Antti Hakanen (Auria Biobank / University of Turku / Hospital District of Southwest Finland, Turku, Finland), Antti Mäkitie (Department of Otorhinolaryngology - Head and Neck Surgery, University of Helsinki and Helsinki University Hospital, Helsinki, Finland), Antti Palomäki (Hospital District of Southwest Finland, Turku, Finland), Anu Loukola (Helsinki Biobank / Helsinki University and Hospital District of Helsinki and Uusimaa, Helsinki), Aoxing Liu (Institute for Molecular Medicine Finland (FIMM), HiLIFE, University of Helsinki, Helsinki, Finland), Apinya Lertratanakul (Abbvie, Chicago, IL, United States), Argyro Bizaki-Vallaskangas (Pirkanmaa Hospital District, Tampere, Finland), Arto Lehisto (Institute for Molecular Medicine Finland (FIMM), HiLIFE, University of Helsinki, Helsinki, Finland), Arto Mannermaa (Biobank of Eastern Finland / University of Eastern Finland / Northern Savo Hospital District, Kuopio, Finland), Athena Matakidou (Astra Zeneca, Cambridge, United Kingdom), Audrey Chu (GlaxoSmithKline, Brentford, United Kingdom), Auli Toivola (THL Biobank / Finnish Institute for Health and Welfare (THL), Helsinki, Finland), Awaisa Ghazal (Institute for Molecular Medicine Finland (FIMM), HiLIFE, University of Helsinki, Helsinki, Finland), Benjamin Challis (Astra Zeneca, Cambridge, United Kingdom), Bridget Riley-Gillis (Abbvie, Chicago, IL, United States), Bridget Riley-Gills (Abbvie, Chicago, IL, United States), Caroline Fox (Merck, Kenilworth, NJ, United States), Chia-Yen Chen (Biogen, Cambridge, MA, United States), Chris O'Donnell (Novartis Institutes for BioMedical Research, Cambridge, MA, United States), Clément Chatelain (Translational Sciences, Sanofi R&D, Framingham, MA, USA), Daniel Gordin (Hospital District of Helsinki and Uusimaa, Helsinki, Finland), David Choy (Genentech, San Francisco, CA, United States), David Pulford (GlaxoSmithKline, Stevenage, United Kingdom), David Rice (Hospital District of Helsinki and Uusimaa, Helsinki, Finland), Dawn Waterworth (Janssen Research & Development, LLC, Spring House, PA, United States), Debby Ngo (Novartis, Basel, Switzerland), Deepak Raipal (Translational Sciences, Sanofi R&D, Framingham, MA, USA), Dermot Reilly (Janssen Research & Development, LLC, Boston, MA, United States), Diptee Kulkarni (GlaxoSmithKline, Brentford, United Kingdom), Dirk Paul (Astra Zeneca, Cambridge, United Kingdom), Edmond Teng (Genentech, San Francisco, CA, United States), Eero Punkka (Helsinki Biobank / Helsinki University and Hospital District of Helsinki and Uusimaa, Helsinki), Eeva Kangasniemi (Finnish Clinical Biobank Tampere / University of Tampere / The Wellbeing Services County of Pirkanmaa, Tampere, Finland), Eeva Sliz (University of Oulu, Oulu, Finland), Eija Laakkonen (University of Jyväskylä, Jyväskylä, Finland), Ekaterina Khramtsova (Janssen Research & Development, LLC, Spring House, PA, United States), Elina Järvensivu (THL Biobank / Finnish Institute for Health and Welfare (THL), Helsinki, Finland), Elina Kilpeläinen (Institute for Molecular Medicine Finland (FIMM), HiLIFE, University of Helsinki, Helsinki, Finland), Elisa Rahikkala (Wellbeing Services County of North Ostrobothnia, Oulu, Finland), Elisabeth Widen (Institute for Molecular Medicine Finland (FIMM), HiLIFE, University of Helsinki, Helsinki, Finland), Elmo Saarentaus (Institute for Molecular Medicine Finland (FIMM), HiLIFE, University of Helsinki, Helsinki, Finland), Eric Green (Maze Therapeutics, San Francisco, CA, United States), Erich Strauss (Genentech, San Francisco, CA, United States), Erkki Isometsä (Hospital District of Helsinki and Uusimaa, Helsinki, Finland), Esa Pitkänen (Institute for Molecular Medicine Finland (FIMM), HiLIFE, University of Helsinki, Helsinki, Finland), Essi Kaiharju (THL Biobank / Finnish Institute for Health and Welfare (THL), Helsinki, Finland), Eveliina Salminen (Hospital District of Helsinki and Uusimaa, Helsinki, Finland), Fabiana Farias (Merck, Kenilworth, NJ, United States), Fanli Xu (GlaxoSmithKline, Brentford, United Kingdom), Fedik Rahimov (Abbvie, Chicago, IL, United States), Felix Vaura (Finnish Institute for Health and Welfare (THL), Helsinki, Finland), Fredrik Åberg (Transplantation and Liver Surgery Clinic, Helsinki University Hospital, Helsinki University, Helsinki, Finland), George Okafo (Boehringer Ingelheim, Ingelheim am Rhein, Germany), Glenda Lassi (Astra Zeneca, Cambridge, United Kingdom), Hanna Ollila (Institute for Molecular Medicine Finland (FIMM), HiLIFE, University of Helsinki, Helsinki, Finland), Hannele Laivuori (Institute for Molecular Medicine Finland (FIMM), HiLIFE, University of Helsinki, Helsinki, Finland), Hannele Mattsson (THL Biobank / Finnish Institute for Health and Welfare (THL), Helsinki, Finland), Hannu Kankaanranta (University of Gothenburg, Gothenburg, Sweden/ Seinäjoki Central Hospital, Seinäjoki, Finland/ Tampere University, Tampere, Finland), Hannu Uusitalo (Pirkanmaa Hospital District, Tampere, Finland), Hao Chen (Genentech, San Francisco, CA, United States), Harri Siirtola (University of Tampere, Tampere, Finland), Heidi Silven (University of Oulu, Oulu, Finland), Heikki Joensuu (Hospital District of Helsinki and Uusimaa, Helsinki, Finland), Heli Lehtonen (Pfizer, New York, NY, United States), Heli Salminen-Mankonen (Boehringer Ingelheim, Ingelheim am Rhein, Germany), Henna Palin (Finnish Clinical Biobank Tampere / University of Tampere / The Wellbeing Services County of Pirkanmaa, Tampere, Finland), Henrike Heyne (Institute for Molecular Medicine Finland (FIMM), HiLIFE, University of Helsinki, Helsinki, Finland), Hilkka Soininen (Northern Savo Hospital District, Kuopio, Finland), Howard Jacob (Abbvie, Chicago, IL, United States), Hubert Chen (Genentech, San Francisco, CA, United States), Huei-Yi Shen (Institute for Molecular Medicine Finland (FIMM), HiLIFE, University of Helsinki, Helsinki, Finland), Iida Vähätalo (University of Tampere, Tampere, Finland), Iiris Hovatta (University of Helsinki, Finland), Ilkka Kalliala (Hospital District of Helsinki and Uusimaa, Helsinki, Finland), Ioanna Tachmazidou (Astra Zeneca, Cambridge, United Kingdom), Jaakko Parkkinen (Pfizer, New York, NY, United States), Jaakko Tyrmi (University of Oulu, Oulu, Finland / University of Tampere, Tampere, Finland), Jaana Suvisaari (Finnish Institute for Health and Welfare (THL), Helsinki, Finland), Jae-Hoon Sul (Merck, Kenilworth, NJ, United States), Janet Kumar (GlaxoSmithKline, Collegeville, PA, United States), Jani Tikkanen (Northern Finland Biobank Borealis / University of Oulu / Wellbeing Services County of North Ostrobothnia, Oulu, Finland), Jari Laukkanen (Central Finland Biobank / University of Jyväskylä / Central Finland Health Care District, Jyväskylä, Finland), Jarmo Ritari (Finnish Red Cross Blood Service, Helsinki, Finland), Jason Miller (Merck, Kenilworth, NJ, United States), Javier Garcia-Tabuenca (University of Tampere, Tampere, Finland), Javier Gracia-Tabuenca (University of Tampere, Tampere, Finland), Jeffrey Waring (Abbvie, Chicago, IL, United States), Jenni Aittokallio (Hospital District of Southwest Finland, Turku, Finland), Jennifer Schutzman (Genentech, San Francisco, CA, United States), Jiwoo Lee (Institute for Molecular Medicine Finland (FIMM), HiLIFE, University of Helsinki, Helsinki, Finland; Broad Institute, Cambridge, MA, United States), Joanna Betts (GlaxoSmithKline, Brentford, United Kingdom), Joel Rämö (Institute for Molecular Medicine Finland (FIMM), HiLIFE, University of Helsinki, Helsinki, Finland), Johanna

Huhtakangas (Department of Internal Medicine, Kuopio University Hospital, Kuopio, Finland), Johanna Mattson (Hospital District of Helsinki and Uusimaa, Helsinki, Finland), Johanna Mäkelä (FINBB - Finnish biobank cooperative), Johanna Schleutker (Auria Biobank / Univ. of Turku / Hospital District of Southwest Finland, Turku, Finland), Johannes Kettunen (Northern Finland Biobank Borealis / University of Oulu / Northern Ostrobothnia Hospital District, Oulu, Finland), John Eicher (GlaxoSmithKline, Brentford, United Kingdom), Joni A Turunen (Helsinki University Hospital and University of Helsinki, Helsinki, Finland; Eye Genetics Group, Folkhälsan Research Center, Helsinki, Finland), Jorge Esparza Gordillo (GlaxoSmithKline, Brentford, United Kingdom), Joseph Maranville (Bristol Myers Squibb, New York, NY, United States), Juha Karjalainen (Institute for Molecular Medicine Finland (FIMM), HiLIFE, University of Helsinki, Helsinki, Finland), Juha Mehtonen (Institute for Molecular Medicine Finland (FIMM), HiLIFE, University of Helsinki, Helsinki, Finland), Juha Rinne (Hospital District of Southwest Finland, Turku, Finland), Juha Sinisalo (Hospital District of Helsinki and Uusimaa, Helsinki, Finland), Jukka Koskela (Hospital District of Helsinki and Uusimaa, Helsinki, Finland), Jukka Partanen (Finnish Red Cross Blood Service / Finnish Hematology Registry and Clinical Biobank, Helsinki, Finland), Jukka Peltola (Pirkanmaa Hospital District, Tampere, Finland), Julie Hunkapiller (Genentech, San Francisco, CA, United States), Jussi Hernesniemi (Pirkanmaa Hospital District, Tampere, Finland), Juulia Partanen (Institute for Molecular Medicine Finland, HiLIFE, University of Helsinki, Finland), Jyrki Pitkänen (Institute for Molecular Medicine Finland (FIMM), HiLIFE, University of Helsinki, Helsinki, Finland), Kai Kaarniranta (Northern Savo Hospital District, Kuopio, Finland), Kaisa Tasanen (Wellbeing Services County of North Ostrobothnia, University of Oulu, Oulu, Finland), Kaj Metsärinne (Hospital District of Southwest Finland, Turku, Finland), Kalle Pärn (Institute for Molecular Medicine Finland (FIMM), HiLIFE, University of Helsinki, Helsinki, Finland), Karen He (Janssen Research & Development, LLC, Spring House, PA, United States), Kari Eklund (Hospital District of Helsinki and Uusimaa, Helsinki, Finland), Katariina Hannula-Jouppi (Hospital District of Helsinki and Uusimaa, Helsinki, Finland), Katherine Klinger (Translational Sciences, Sanofi R&D, Framingham, MA, USA), Kati Donner (Institute for Molecular Medicine Finland (FIMM), HiLIFE, University of Helsinki, Helsinki, Finland), Kati Hyvärinen (Finnish Red Cross Blood Service, Helsinki, Finland), Kati Kristiansson (THL Biobank / Finnish Institute for Health and Welfare (THL), Helsinki, Finland), Katja Kivinen (Institute for Molecular Medicine Finland (FIMM), HiLIFE, University of Helsinki, Helsinki, Finland), Katri Kaukinen (Pirkanmaa Hospital District, Tampere, Finland), Katri Pylkäs (University of Oulu, Oulu, Finland), Katriina Aalto-Setälä (Faculty of Medicine and Health Technology, Tampere University, Tampere, Finland), Kimmo Palin (University of Helsinki, Helsinki, Finland), Kirsi Auro (GlaxoSmithKline, Espoo, Finland), Kirsi Kalpala (Pfizer, New York, NY, United States), Kirsi Sipilä (Research Unit of Oral Health Sciences Faculty of Medicine, University of Oulu, Oulu, Finland; Medical Research Center, Oulu, Oulu University Hospital and University of Oulu, Oulu, Finland), Klaus Elenius (Hospital District of Southwest Finland, Turku, Finland), Kristiina Aittomäki (Department of Medical Genetics, Helsinki University Central Hospital, Helsinki, Finland), Kristin Tsuo (Institute for Molecular Medicine Finland (FIMM), HiLIFE, University of Helsinki, Helsinki, Finland; Broad Institute, Cambridge, MA, United States), L. Elisa Lahtela (Institute for Molecular Medicine Finland (FIMM), HiLIFE, University of Helsinki, Helsinki, Finland), Laura Addis (GlaxoSmithKline, Brentford, United Kingdom), Laura Huilaja (Wellbeing Services County of North Ostrobothnia, University of Oulu, Oulu, Finland), Laura Kotaniemi-Talonen (Pirkanmaa Hospital District, Tampere, Finland), Laura Pirilä (Hospital District of Southwest Finland, Turku, Finland), Laure Morin-Papunen (Wellbeing Services County of North Ostrobothnia, Oulu, Finland), Lauri Aaltonen (Hospital District of Helsinki and Uusimaa, Helsinki, Finland), Leena Koulu (Hospital District of Southwest Finland, Turku, Finland), Liisa Suominen (Northern Savo Hospital District, Kuopio, Finland), Linda McCarthy (GlaxoSmithKline, Brentford, United Kingdom), Lotta Männikkö (THL Biobank / Finnish Institute for Health and Welfare (THL), Helsinki, Finland), Ma'en Obeidat (Novartis Institutes for BioMedical Research, Cambridge, MA, United States), Maarit Niinimäki (Wellbeing Services County of North Ostrobothnia, Oulu, Finland), Majd Mouded (Novartis, Basel, Switzerland), Malla-Maria Linna (Helsinki Biobank / Helsinki University and Hospital District of Helsinki and Uusimaa, Helsinki), Manuel Rivas (University of Stanford, Stanford, CA, United States), Marc Jung (Boehringer Ingelheim, Ingelheim am Rhein, Germany), Marco Hautalahti (FINBB - Finnish biobank cooperative), Margaret G. Ehm (GlaxoSmithKline, Collegeville, PA, United States), Margit Pelkonen (Northern Savo Hospital District, Kuopio, Finland), Mari E. Niemi (Institute for Molecular Medicine Finland (FIMM), HiLIFE, University of Helsinki, Helsinki, Finland), Mari Kaunisto (Institute for Molecular Medicine Finland (FIMM), HiLIFE, University of Helsinki, Helsinki, Finland), Maria Siponen (Northern Savo Hospital District, Kuopio, Finland), Marianna Niemi (University of Tampere, Tampere, Finland), Marja Vääräsmäki (Wellbeing Services County of North Ostrobothnia, Oulu, Finland), Marja-Riitta Taskinen (Hospital District of Helsinki and Uusimaa, Helsinki, Finland), Mark Daly (Institute for Molecular Medicine Finland (FIMM), HiLIFE, University of Helsinki, Helsinki, Finland; Broad Institute of MIT and Harvard; Massachusetts General Hospital), Mark McCarthy (Genentech, San Francisco, CA, United States), Markku Laukkanen (THL Biobank / Finnish Institute for Health and Welfare (THL), Helsinki, Finland), Markku Voutilainen (Hospital District of Southwest Finland, Turku, Finland), Markus Perola (THL Biobank / Finnish Institute for Health and Welfare (THL), Helsinki, Finland), Marla Hochfeld (Bristol Myers Squibb, New York, NY, United States), Mart Kals (Institute for Molecular Medicine Finland (FIMM), HiLIFE, University of Helsinki, Helsinki, Finland), Martti Färkkilä (Hospital District of Helsinki and Uusimaa, Helsinki, Finland), Mary Pat Reeve (Institute for Molecular Medicine Finland (FIMM), HiLIFE, University of Helsinki, Helsinki, Finland), Masahiro Kanai (Broad Institute, Cambridge, MA, United States), Meijian Guan (Janssen Research & Development, LLC, Spring House, PA, United States), Melissa Miller (Pfizer, New York, NY, United States), Mengzhen Liu (Abbvie, Chicago, IL, United States), Mervi Aavikko (Institute for Molecular Medicine Finland (FIMM), HiLIFE, University of Helsinki, Helsinki, Finland), Mika Helminen (University of Tampere, Tampere, Finland), Mika Kähönen (Finnish Clinical Biobank Tampere / University of Tampere / Pirkanmaa Hospital District, Tampere, Finland), Mike Mendelson (Novartis, Boston, MA, United States), Mikko Arvas (Finnish Red Cross Blood Service / Finnish Hematology Registry and Clinical Biobank, Helsinki, Finland), Mikko Kiviniemi (Northern Savo Hospital District, Kuopio, Finland), Minna Brunfeldt (THL Biobank / Finnish Institute for Health and Welfare (THL), Helsinki, Finland), Minna Karjalainen (University of Oulu, Oulu, Finland), Minna Raivio (Hospital District of Helsinki and Uusimaa, Helsinki, Finland), Mitja Kurki (Institute for Molecular Medicine Finland (FIMM), HiLIFE, University of Helsinki, Helsinki, Finland; Broad Institute, Cambridge, MA, United States), Mutaamba Maasha (Broad Institute, Cambridge, MA, United States), Nan Bing (Pfizer, New York, NY, United States), Natalia Pujol (Estonian biobank, Tartu, Estonia), Natalie Bowers (Genentech, San Francisco, CA, United States), Nathan Lawless (Boehringer Ingelheim, Ingelheim am Rhein, Germany), Neha Raghavan (Merck, Kenilworth, NJ, United States), Nicole Renaud (Novartis Institutes for BioMedical Research, Cambridge, MA, United States), Niko Välimäki (University of Helsinki, Helsinki, Finland), Nina Mars (Institute for Molecular Medicine Finland (FIMM), HiLIFE, University of Helsinki, Helsinki, Finland), Nina Pitkänen (Auria Biobank / University of Turku / Hospital District of Southwest Finland, Turku, Finland), Nizar Smaoui (Abbvie, Chicago, IL, United States), Oili Kaipiainen-Seppänen (Northern Savo Hospital District, Kuopio, Finland), Olli Carpén (Helsinki Biobank / Helsinki University and Hospital District of Helsinki and Uusimaa, Helsinki), Oluwaseun Alexander Dada (Institute for Molecular Medicine Finland (FIMM), HiLIFE, University of Helsinki, Helsinki, Finland), Oskari Heikinheimo (Hospital District of Helsinki and Uusimaa, Helsinki, Finland), Outi Tuovila (Business Finland, Helsinki, Finland), Outi Uimari (University of Oulu, Wellbeing Services County of North Ostrobothnia, Oulu, Finland), Paula Kauppi (Hospital District of Helsinki and Uusimaa, Helsinki, Finland), Peeter Karihtala (Helsinki University Hospital Comprehensive Cancer Centre, Helsinki, Finland), Pekka Nieminen (Hospital District of Helsinki and Uusimaa, Helsinki, Finland), Pentti Tienari (Hospital District of Helsinki and Uusimaa, Helsinki, Finland), Perttu Terho (Auria Biobank / University of Turku / Hospital District of Southwest Finland, Turku, Finland), Petri Virolainen (Auria Biobank / University of Turku / Hospital District of Southwest Finland, Turku, Finland), Pia Isomäki (Pirkanmaa Hospital District, Tampere, Finland), Pietro Della Briotta Parolo (Institute for Molecular Medicine Finland (FIMM), HiLIFE, University of Helsinki, Helsinki, Finland), Pirkko Pussinen (Hospital District of Helsinki and Uusimaa, Helsinki, Finland), Priit Palta (Institute for Molecular Medicine Finland (FIMM), HiLIFE, University of Helsinki, Helsinki, Finland), Päivi Auvinen (Northern Savo Hospital District, Kuopio, Finland), Päivi Laiho (THL Biobank / Finnish Institute for Health and Welfare (THL), Helsinki, Finland), Päivi Mäntylä (Northern Savo Hospital District, Kuopio, Finland), Qingqin S Li (Janssen Research & Development, LLC, Titusville, NJ 08560, United States), Raimo Pakkanen (Business Finland, Helsinki, Finland), Rajashree Mishra (GlaxoSmithKline, Brentford, United Kingdom), Reetta Hinttala (Northern Finland Biobank Borealis / University of Oulu / Northern Ostrobothnia Hospital District, Oulu, Finland), Reetta Kälviäinen (Northern Savo Hospital District, Kuopio, Finland), Regis Wong (THL Biobank / Finnish Institute for Health and Welfare (THL), Helsinki, Finland), Relja Popovic (Abbvie, Chicago, IL, United States), Rigbe Weldatsadik (Institute for Molecular Medicine Finland (FIMM), HiLIFE, University of Helsinki, Helsinki, Finland), Riikka Arffman (University of Oulu, Oulu, Finland), Riitta Lahesmaa (Hospital District of Southwest Finland, Turku, Finland), Rion Pendergrass (Genentech, San Francisco, CA, United States), Rion Pendergrass (Genentech, San Francisco, CA, United States), Risto Kajanne (Institute for Molecular Medicine Finland (FIMM), HiLIFE, University of Helsinki, Helsinki, Finland), Robert Graham (Maze Therapeutics, San Francisco, CA, United States), Robert Plenge (Bristol Myers Squibb, New York, NY, United States), Robert Yang (Janssen Biotech, Beerse, Belgium), Rodos Rodosthenous (Institute for Molecular Medicine Finland (FIMM), HiLIFE, University of Helsinki, Helsinki, Finland), Roosa

Kallionpää (Hospital District of Southwest Finland, Turku, Finland), Sahar Mozaffari (Maze Therapeutics, San Francisco, CA, United States), Sally John (Biogen, Cambridge, MA, United States), Sami Heikkinen (University of Eastern Finland, Kuopio, Finland), Sami Koskelainen (THL Biobank / Finnish Institute for Health and Welfare (THL), Helsinki, Finland), Sampsa Pikkarainen (Hospital District of Helsinki and Uusimaa, Helsinki, Finland), Samuel Lessard (Translational Sciences, Sanofi R&D, Framingham, MA, USA), Samuli Ripatti (Institute for Molecular Medicine Finland (FIMM), HiLIFE, University of Helsinki, Helsinki, Finland), Sanna Toppila-Salmi (University of Helsinki, Helsinki, Finland), Sanni Lahdenperä (Biogen, Cambridge, MA, United States), Sanni Ruotsalainen (Institute for Molecular Medicine Finland (FIMM), HiLIFE, University of Helsinki, Helsinki, Finland), Sarah Smith (Finnish Biobank Cooperative - FINBB), Satu Strausz (Institute for Molecular Medicine Finland (FIMM), HiLIFE, University of Helsinki, Helsinki, Finland), Sauli Vuoti (Janssen-Cilag Oy, Espoo, Finland), Shabbeer Hassan (Institute for Molecular Medicine Finland (FIMM), HiLIFE, University of Helsinki, Helsinki, Finland), Shameek Biswas (Bristol Myers Squibb, New York, NY, United States), Shanmukha Sampath Padmanabhuni (Institute for Molecular Medicine Finland (FIMM), HiLIFE, University of Helsinki, Helsinki, Finland), Shuang Luo (Institute for Molecular Medicine Finland (FIMM), HiLIFE, University of Helsinki, Helsinki, Finland), Simonne Longerich (Merck, Kenilworth, NJ, United States), Sini Lähteenmäki (THL Biobank / Finnish Institute for Health and Welfare (THL), Helsinki, Finland), Sirkku Peltonen (Hospital District of Southwest Finland, Turku, Finland), Sirpa Soini (THL Biobank / Finnish Institute for Health and Welfare (THL), Helsinki, Finland), Stefan McDonough (Pfizer, New York, NY, United States), Stephanie Loomis (Biogen, Cambridge, MA, United States), Susan Eaton (Biogen, Cambridge, MA, United States), Susanna Lemmelä (Institute for Molecular Medicine Finland (FIMM), HiLIFE, University of Helsinki, Helsinki, Finland), Susanna Savukoski (University of Oulu, Oulu, Finland), Taneli Raivio (Helsinki Biobank / Helsinki University and Hospital District of Helsinki and Uusimaa, Helsinki), Tarja Laitinen (Finnish Clinical Biobank Tampere / University of Tampere / The Wellbeing Services County of Pirkanmaa, Tampere, Finland), Taru Tukiainen (Institute for Molecular Medicine Finland (FIMM), HiLIFE, University of Helsinki, Helsinki, Finland), Teea Salmi (Pirkanmaa Hospital District, Tampere, Finland), Teemu Niiranen (Finnish Institute for Health and Welfare (THL), Helsinki, Finland), Teemu Paajanen (THL Biobank / Finnish Institute for Health and Welfare (THL), Helsinki, Finland), Terhi Kilpi (THL Biobank / Finnish Institute for Health and Welfare (THL), Helsinki, Finland), Terhi Ollila (Hospital District of Helsinki and Uusimaa, Helsinki, Finland), Terhi Piltonen (University of Oulu, Wellbeing Services County of North Ostrobothnia, Oulu, Finland), Tero Hiekkalinna (THL Biobank / Finnish Institute for Health and Welfare (THL), Helsinki, Finland), Terttu Harju (Northern Ostrobothnia Hospital District, Oulu, Finland), Thomas Damm Als (Aarhus University, Denmark), Tiina Luukkaala (University of Tampere, Tampere, Finland), Tiinamaija Tuomi (Hospital District of Helsinki and Uusimaa, Helsinki, Finland), Tim Lu (Genentech, San Francisco, CA, United States), Timo Blomster (Wellbeing Services county of North Ostrobothnia, Oulu, Finland), Timo Hiltunen (Hospital District of Helsinki and Uusimaa, Helsinki, Finland), Timo P. Sipilä (Institute for Molecular Medicine Finland (FIMM), HiLIFE, University of Helsinki, Helsinki, Finland), Tom Southerington (Finnish Biobank Cooperative - FINBB), Tomi P. Mäkelä (HiLIFE, University of Helsinki, Finland, Finland), Triin Laisk (Estonian biobank, Tartu, Estonia), Tuomo Kiiskinen (Institute for Molecular Medicine Finland (FIMM), HiLIFE, University of Helsinki, Helsinki, Finland), Tuomo Mantere (Northern Finland Biobank Borealis / University of Oulu / Wellbeing Services County of North Ostrobothnia, Oulu, Finland), Tuomo Meretoja (Hospital District of Helsinki and Uusimaa, Helsinki, Finland), Tuula Palotie (University of Helsinki and Hospital District of Helsinki and Uusimaa, Helsinki, Finland), Tuula Salo (Hospital District of Helsinki and Uusimaa, Helsinki, Finland), Tuuli Sistonen (THL Biobank / Finnish Institute for Health and Welfare (THL), Helsinki, Finland), Tytti Willberg (Hospital District of Southwest Finland, Turku, Finland), Ulla Palotie (Hospital District of Helsinki and Uusimaa, Helsinki, Finland), Ulvi Gursoy (Hospital District of Southwest Finland, Turku, Finland), Varpu Jokimaa (Hospital District of Southwest Finland, Turku, Finland), Veikko Salomaa (Finnish Institute for Health and Welfare (THL), Helsinki, Finland), Veli-Matti Kosma (Biobank of Eastern Finland / University of Eastern Finland / Northern Savo Hospital District, Kuopio, Finland), Venla Kurra (Pirkanmaa Hospital District, Tampere, Finland), Vincent Llorens (Institute for Molecular Medicine Finland (FIMM), HiLIFE, University of Helsinki, Helsinki, Finland), Vuokko Anttonen (Wellbeing Services County of North Ostrobothnia, University of Oulu, Oulu, Finland), Wei Zhou (Broad Institute, Cambridge, MA, United States), Xinli Hu (Pfizer, New York, NY, United States), Ying Wu (Pfizer, New York, NY, United States), Zhihao Ding (Boehringer Ingelheim, Ingelheim am Rhein, Germany).

**Contributors** TTS: writing the original draft, revision and data collection. EV: writing the original draft, revision, resources, concept, design and supervision. AA: writing, revision and data collection. IR, HL, S-KH, SK, TK, SA: writing. MO, P-IM: revision. AK: design, administration and revision. AT: study coordination and revision. TV, HV-Y, MM, RS, PN, SS: resources and revision. V-MK, TK, EP, MR: resources. JK: concept, design, revision and resources. MU: study coordination, data collection. VJ: concept, design, revision, supervision. MMF, MH, JL, RK: consultation in the design of the study. HR: concept, design, revision. AP: concept, resources, revision. TP: data management. AJ: administration. FinnGen: administration, resources, funding. TS and EV contributed equally to this paper.

**Funding** TWINGEN study was funded by the FinnGen project with the aim to enrich the phenotype information in FinnGen to achieve the goals of the project. Finnish Twin Cohort study has been supported by the Academy of Finland (Grants 265240, 263278, 308248), the Sigrid Jusélius Foundation, NIH/NHLBI grant HL104125 and NIH Grant R01 AG060470. EV was supported by the Academy of Finland (grants 314639 and 345988). JK was supported by Centre of Excellence in Complex Disease Genetics (grants #312073 and #336823 from the Academy of Finland). The FinnGen project is funded by Business Finland and AbbVie, AstraZeneca UK, Biogen, Bristol Myers Squibb (and Celgene Corporation and Celgene International II), Genentech, Merck Sharp and Dohme LLC, a subsidiary of Merck & Co, Rahway, NJ, USA, Pfizer, GlaxoSmithKline Intellectual Property Development, Sanofi US Services, Maze Therapeutics, Janssen Biotech, Novartis, and Boehringer Ingelheim. All Finnish biobanks are members of the BBMRI.fi infrastructure (https://www.bbmri.fi). The FINBB (https://finbb.fi/) is the coordinator of BBMRI-ERIC operations in Finland. The Finnish biobank data can be accessed through the Fingenious services (https://site.fingenious.fi/en/) managed by FINBB.

**Competing interests** AP is the Chief Scientific Officer of the FinnGen project that is funded by 13 pharmaceutical companies. HR was a full-time employee of Biogen during study planning and manuscript drafting and has stocks at Merck & Co and Biogen. MMF has received development funding from the Regional Council of Northern Savo and Business Finland for a data-driven tool related to memory disorders and healthcare decision tools, Charles River DRS Finland Ltd. and Orion Pharma have donated equipment for nonclinical cognition testing at the University of Eastern Finland. PI-M has received funding from Orion Research Foundation and Helsinki Biomedicum Foundation outside the present work. RK declares funding paid to the institution from Academy of Finland, 27 Government research funding, Saastamoinen Foundation, Vaajasalo foundation and Jane and Aatos Erkko foundation outside the present work; consulting fees from Orion Pharma; payment or honoraria from Angelini Pharma, Jazz Pharma, Lundbeck, Eisai, Orion Pharma, OmaMedical, Takeda, UCB; participation in monitoring or advisory board from Marinus Pharma and UCB; and leadership or fiduciary role in European Academy of Neurology Epilepsy scientific panel management group, European Epilepsy Reference network Epicare Steering Group and International League Against Epilepsy Career Development Commission. RS declares stocks or stock options at Orion Pharma. S-KH declares payment or honoraria and support for attending meetings or travel from Roche, consulting fees and participation in monitoring or advisory board from Novartis. TK declares payment or honoraria from Novartis Finland and Bayer Nordic SE. The authors declare no other competing financial or non-financial Interests.

**Patient and public involvement** Patients and/or the public were not involved in the design, or conduct, or reporting, or dissemination plans of this research.

**Patient consent for publication** Not applicable.

**Provenance and peer review** Not commissioned; externally peer reviewed.

**ORCID iDs**
Eero Vuoksimaa http://orcid.org/0000-0002-6534-3667
Toni T Saari http://orcid.org/0000-0001-7721-3336
Sari Aaltonen http://orcid.org/0000-0002-2873-4263

Sanna-Kaisa Herukka http://orcid.org/0000-0003-3308-8662
Paula Iso-Markku http://orcid.org/0000-0001-6683-919X
Tarja Kokkola http://orcid.org/0000-0002-3303-3912
Tommi Vasankari http://orcid.org/0000-0001-7209-9351
Mikko Hiltunen http://orcid.org/0000-0003-3566-4096
Reetta Kälviäinen http://orcid.org/0000-0003-2935-5131
Minna Männikkö http://orcid.org/0000-0002-6857-6423
Aarno Palotie http://orcid.org/0000-0002-2527-5874
Jaakko Kaprio http://orcid.org/0000-0002-3716-2455
Heiko Runz http://orcid.org/0000-0002-2133-7345

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
