## [Reviewer comments · BMJ Open]

ARTICLE DETAILS

TITLE (PROVISIONAL)	TWINGEN – protocol for an observational clinical biobank recall and biomarker cohort study to identify Finnish individuals with high risk of Alzheimer’s disease
AUTHORS	Saari, Toni; Vuoksimaa, Eero; Aaltonen, Aino; Aaltonen, Sari; Herukka, Sanna-Kaisa; Iso-Markku, Paula; Kokkola, Tarja; Kyttälä, Aija; Kärkkäinen, Sari; Liedes, Hilikka; Ollikainen, Miina; Palviainen, Teemu; Ruotsalainen, Ilona; Toivola, Auli; Urjansson, Mia; Vasankari, Tommi; Vähä-Ypyä, Henri; Forsberg, Markus M.; Hiltunen, Mikko; Jalanko, Anu; Kälviäinen, Reetta; Kuopio, Teijo; Lähteenmäki, Jaakko; Nyberg, Pia; Männikkö, Minna; Serpi, Raisa; Siltanen, Sanna; FinnGen, -; Palotie, Aarno; Kaprio, Jaakko; Runz, Heiko; Julkunen, Valterti

VERSION 1 – REVIEW

REVIEWER	Shirzadi, Zahra Harvard University
REVIEW RETURNED	17-Dec-2023

GENERAL COMMENTS	The manuscript entitled “TWINGEN – protocol for an observational clinical biobank recall and biomarker study to identify individuals with a high risk of Alzheimer’s disease” described a protocol that is used to examine the feasibility of using a biobank recall strategy to identify people at risk of AD in the general population. The study is interesting, multi-modal, and feasible. Please see my comments below. 1. The study approach is not very clear. I read the article a few times, but I am still unsure which biobank datasets were used as parents for this study. Please revise the text and clarify a) which datasets you used and b) why you chose those datasets.2. Would it be possible to make a figure similar to Figure 1 for the sub-studies you aim to perform?3. Also, please state how the AD diagnosis will be done in your study. This study aims to identify people at risk of AD; how will you determine that? Is it through plasma markers or cognitive tests?4. Please mention why you chose a sample size = 50 for the physical activity assessment. Is it based on a power analysis or feasibility?
---

REVIEWER	Amano, Takashi Rutgers University Newark
REVIEW RETURNED	10-Jan-2024

GENERAL COMMENTS	We extend our gratitude to the authors for their invaluable manuscript outlining the protocol for an observational clinical biobank recall and biomarker study aimed at identifying individuals at high risk of Alzheimer’s disease (AD). The manuscript delineates the protocol for a study in Finland that will gather data from participants of TWINGEN, a population-based follow-up study investigating the feasibility of easily implementable methods for assessing AD risk. A notable concern pertains to the insufficient elaboration on the analytical methodologies employed in this study. The section labeled “Aims, data analysis, and future directions” lacks the requisite detail. For instance, on page 10, the authors indicate, “By combining all data, we aim to stratify our participants into sub-groups of low, intermediate, and high AD risk based on genetic, biomarker, cognitive, lifestyle, and symptom data” without specifying the methodologies employed. Furthermore, clarification is needed regarding phrases like “correlation analysis,” “calculation of factor scores,” and the utilization of these factors in predicting cognitive and biomarker status. Additionally, on page 11, the authors mention, “Additional parameters of physical activity and sleep are also available for detailed analyses.” A more explicit elucidation of this statement is warranted. Furthermore, it would be beneficial to discuss limitations within the main body of the text rather than confining them solely to the article summary.
--

VERSION 1 – AUTHOR RESPONSE

Reviewer: 1

Dr. Zahra Shirzadi, Harvard University

Comments to the Author:

The manuscript entitled “TWINGEN – protocol for an observational clinical biobank recall and biomarker study to identify individuals with a high risk of Alzheimer’s disease” described a protocol that is used to examine the feasibility of using a biobank recall strategy to identify people at risk of AD in the general population. The study is interesting, multi-modal, and feasible.

Please see my comments below.

1. The study approach is not very clear. I read the article a few times, but I am still unsure which biobank datasets were used as parents for this study. Please revise the text and clarify a) which datasets you used and b) why you chose those datasets.

Response: We have now clarified the selection and justification of the datasets. In brief, the participants are older Finnish Twin Cohort (FTC) study participants, whose samples and data have been transferred to THL Biobank in 2018. FTC participants were chosen as they have extensive longitudinal data starting from 1975 that can be combined with the newly collected data in TWINGEN.

For the purposes of this study, eligible FTC participants have been screened by THL Biobank and the data collection has been carried out by the FTC team at the University of Helsinki and in five biobanks throughout Finland and Turku University of Applied Sciences. The names of the biobanks have been

listed in the manuscript. We have made some clarifications regarding the selection of study participants.

Abstract (p. 3): *“The data collected in TWINGEN will be returned to THL Biobank from where it can later be requested for other biobank studies such as FinnGen that supported TWINGEN.”*

The subheading *THL Biobank and the FinnGen study* and its contents have been removed as the information in this paragraph was partially redundant with *Study participant selection* and *Data collection*, and the remaining information is better presented elsewhere in the manuscript:

Study participant selection (p. 6–7): *“The FTC cohort was chosen as the primary target of this study because it is a population-based follow-up study with up to 48 years of previous comprehensive health data available. Combining the historic data with newly collected samples would allow building of longitudinal trajectories of various lifestyle and health factors to late-life cognitive decline. The main selection criteria in the biobank were previous participation in FTC, age (65–85), place of current residence in Finland, Finnish as the first language, and no known diagnosis affecting cognition in biobank records. The selection of eligible FTC study participants was done through THL Biobank and data collection was carried out by the University of Helsinki for those living in greater Helsinki area or surrounding regions and by regional biobanks and Turku University of Applied Science based on the residency of the participant. Selection also included participation in the FinnGen nationwide biobank research study for two reasons.⁸ First, FinnGen supported the collection of the TWINGEN cohort with the aim to enrich phenotype information. Second, the collected TWINGEN data are to be returned to THL Biobank, from where they can later be requested for the FinnGen study and combined with its extensive gene and health register data. Thus far, FinnGen has produced genotype data from ca. 500 000 biobank donors of all Finnish biobanks to perform large scale genome and health research. Below, we describe each of the data sources and the study protocol of TWINGEN.”*

Additionally, the role of THL Biobank has been emphasized throughout the text, and we have now used “THL Biobank” instead of “the THL Biobank” consistently.

Abstract (p. 3): *“Suitable candidates were identified and invited to participate in the new study among THL Biobank donors according to TWINGEN study criteria.”*

Abstract (p. 3): *“This recall study consists of FTC/THL Biobank/FinnGen participants whose data were acquired in accordance with the Finnish Biobank Act.”*

Study participant selection (p. 6): *“The target group of the TWINGEN study are individuals who have participated in the older Finnish Twin Cohort (FTC) study of the University of Helsinki (UH), and whose samples and data have been transferred to THL Biobank in 2018.”*

2. Would it be possible to make a figure similar to Figure 1 for the sub-studies you aim to perform?

Response: This is a great idea from the reviewer. We have added Supplementary Figure 2 that clarifies the planned sub-studies. The new figure has been referred to in the opening sentence of Aims, data analysis and future directions.

Aims, data analysis and future directions (p. 22): *“In addition to the overarching aim of assessing the feasibility of biobank recall in the context of preclinical AD, we also have more focused research questions (see Supplementary Figure 2 for an overview of aims and associated data).”*

3. Also, please state how the AD diagnosis will be done in your study. This study aims to identify people at risk of AD; how will you determine that? Is it through plasma markers or cognitive tests?

Response: We have now clarified the diagnostic situation in *Aims, data analysis and future directions*. First, we will follow the participants by using registry-based information. This will allow us to identify those individuals who will develop AD after participation in our study even if they would not participate for later assessments. Secondly, possible follow-up assessment can include gold-standard measures for establishing AD diagnoses.

Aims, data analysis and future directions (p. 10): “*In the initial stage of TWINGEN, we cannot make clinical or research diagnoses of AD, however, we will be able to follow these individuals through national health registry information that are compiled in FinnGen. Registry-based data will allow to predict progression to AD. Possible follow-up visits can also include gold standard measures for diagnosing AD, such as cerebrospinal fluid or positron emission tomography imaging.*”

4. Please mention why you chose a sample size = 50 for the physical activity assessment. Is it based on a power analysis or feasibility?

Response: The justification for the sample size for this part of the study was based on feasibility considerations.

Oura-measured physical activity, sedentary behavior and sleep (p. 20): “*The number of participants asked to wear an Oura ring was an experimental pilot within our larger study. The target sample of 50 was purposed to evaluate the feasibility of a measurement requiring a smart phone app in older adults.*”

Reviewer: 2

Dr. Takashi Amano, Rutgers University Newark

Comments to the Author:

We extend our gratitude to the authors for their invaluable manuscript outlining the protocol for an observational clinical biobank recall and biomarker study aimed at identifying individuals at high risk of Alzheimer’s disease (AD). The manuscript delineates the protocol for a study in Finland that will gather data from participants of TWINGEN, a population-based follow-up study investigating the feasibility of easily implementable methods for assessing AD risk.

1. A notable concern pertains to the insufficient elaboration on the analytical methodologies employed in this study. The section labeled “Aims, data analysis, and future directions” lacks the requisite detail. For instance, on page 10, the authors indicate, “By combining all data, we aim to stratify our participants into sub-groups of low, intermediate, and high AD risk based on genetic, biomarker, cognitive, lifestyle, and symptom data” without specifying the methodologies employed.

Response: The stratification approach has now been explained in more detail with the addition of references for blood-based biomarker cut-offs (Ashton et al., 2024) and for neuropsychological criteria for mild cognitive impairment (Bondi et al., 2014; Vuoksimaa et al., 2020). The opening sentence of the paragraph following the stratification approaches has been slightly modified to flow with the previous text.

Aims, data analysis and future directions (p. 22–23): “*The stratification will be based on a combination of percentiles or cut-offs (e.g., Ashton et al.⁵³) for blood-based biomarkers, cut-offs for cognitive impairment in CERAD,²⁶ cCOG²² and TELE/TICS,¹⁶ lifestyle risk scores and subjective memory complaints. We also aim to use neuropsychological criteria for mild cognitive impairment classification where -1 SD performance in at least two tests are required independent of subjective memory complaints.^{54,55} Biomarkers and cognitive data allow to derive sub-groups based on biomarker profile and cognitive status included in the AT(N) framework.⁵ Additionally, we will use APOE status and PRS for genetic risk profiling although genetics are not included in the AT(N) framework.*”

The stratification approaches are...

2. Furthermore, clarification is needed regarding phrases like “correlation analysis,” “calculation of factor scores,” and the utilization of these factors in predicting cognitive and biomarker status. Additionally, on page 11, the authors mention, “Additional parameters of physical activity and sleep are also available for detailed analyses.” A more explicit elucidation of this statement is warranted.

Response: We have clarified this part of the text for correlation analysis:

Aims, data analysis and future directions (p. 23): *“We aim to assess the comparability of in-person, telephone-based and computerized cognitive assessments using correlation analysis of total scores and tests of different modalities assessing the same cognitive domains (e.g., memory).”*

and for factor scores:

Aims, data analysis and future directions (p. 23): *“The factor scores will be based on exploratory factor analyses of the in-person neuropsychological battery and used instead of individual tests when the interest is on a cognitive domain, not a single test.”*

As for the additional parameters of physical activity and sleep, we intend to convey that the list of parameters in Table 4 is not exhaustive and that a significant number of parameters are available for Oura and the UKK42 accelerometer. The listing of all possible parameters is outside the scope of this protocol. This sentence relating to sleep and physical activity has now been changed.

Aims, data analysis and future directions (p. 23): *“The list of physical activity and sleep parameters in Table 4 is not exhaustive and more parameters are available for detailed analyses.”*

3. Furthermore, it would be beneficial to discuss limitations within the main body of the text rather than confining them solely to the article summary.

Response: Thank you for pointing out that we had included the limitations only in the article summary. We have now included limitations in the discussion section as well.

Discussion (p. 24): *“Although this study has many strengths, a limitation is the lack of gold standard biomarkers (cerebrospinal fluid, positron emission tomography imaging) and neurological examinations in our baseline assessment.”*

Additional remarks from the authors:

1. During the revision process, we noticed that Table 2 had some erroneous information regarding availability of certain data at different timepoints. The following changes have been made:

- The addition of physical activity to year 2011
- The addition of blood pressure to year 1975
- The removal of dietary habits from 2011
- Additionally, we added a row for cholesterol with data available from 1981, 1990, and 2011

Accordingly, Supplementary Table 2 has been revised to include a row for cholesterol. We also added a column in Supplementary Table 2 for the PubMed IDs of the main references mentioned in Table 2, and mentioned that the references are available in Supplementary Table 2.

2. The age range (65–85) has been corrected in two places.

Accelerometer-based physical activity, sedentary behavior and sleep (p. 19): *“Thus, the UKK RM42 accelerometer is usable in the TWINGEN sample of 65–85-year-olds...”*

Study participant selection (p. 7): *“The main selection criteria in the biobank were previous participation in FTC, age (65-85)...”*

3. We have also corrected some affiliations:

- Minerva Foundation Institute for Medical Research, Helsinki, Finland has been added for Miina Ollikainen as her second affiliation
- The affiliation for Aija Kyttälä and Auli Toivola has been changed to include THL Biobank: “THL Biobank, Finnish Institute for Health and Welfare, Helsinki, Finland”
- The affiliation for Teijo Kuopio has been corrected: “Central Finland Biobank, Wellbeing Services County of Central Finland and University of Jyväskylä, Jyväskylä, Finland”

4. The location of Combinostics has been added to be in line with the other companies mentioned in the manuscript.

Computerized web-based cognitive testing (p. 16): *“Web-based cCOG tool created by Combinostics (Tampere, Finland) is used for computerized cognitive testing.”*²²

5. Study participants are now thanked in the acknowledgements.

Acknowledgements (p. 25): *“The participants of the TWINGEN study were recruited through THL Biobank (study number THLBB2022_83) and we thank all study participants for their generous participation in biobank.”*

6. The affiliations in FinnGen collaborator list have been corrected for Raisa Serpi and Jani Tikkanen.

FinnGen Collaborators (p. 36):

Jani Tikkanen	Northern Finland Biobank Borealis / University of Oulu / Wellbeing Services County of North Ostrobothnia, Oulu, Finland
Raisa Serpi	Northern Finland Biobank Borealis / University of Oulu / Wellbeing Services County of North Ostrobothnia, Oulu, Finland

7. Page numbers have been corrected.

VERSION 2 – REVIEW

REVIEWER	Amano, Takashi Rutgers University Newark
REVIEW RETURNED	26-Mar-2024

GENERAL COMMENTS

Thanks to the authors for their sincere and thorough responses to my comments. All of my concerns have been resolved now and I believe this manuscript has a good quality for publication now.